# The autonomy paradox in AI-generated content adoption: Creative-specific alternative to TAM model in China's micro-short drama industry

**Chao Tang***

School of Animation, Jinling Institute of Technology, Nanjing, Jiangsu, China

* tangchao726@jit.edu.cn

## Abstract

In China's booming micro-short drama industry, Artificial Intelligence Generated Content (AIGC) presents creators with an 'autonomy paradox': improving efficiency while sparking fears of lost control, amplified by collectivist culture that heightens tensions between AI-driven productivity and loss of autonomy. Based on a mixed-methods study of 607 micro-short drama creators, this research proposes and tests the Creative Industries Technology Acceptance Model (CITAM), which builds upon TAM foundations while adapting constructs for creative contexts to reveal key dynamics in adoption intentions. Building upon TAM theoretical foundations while introducing innovation compatibility (IC) and creative autonomy retention (CAR), CITAM is grounded in Diffusion of Innovations and Self-Determination Theory to address both rational and psychological adoption factors in creative contexts. Using a mixed methods approach with SEM in 607 surveys and 10 in-depth interviews, the results reveal that CAR positively influences AIGC adoption through IC as a mediator, while CAR negatively moderates the positive influence of IC on adoption intentions, highlighting a modest but significant psychological tension. Qualitative insights on 'Creativity Amplification' complement this, showing that creators perceive AIGC as an idea enhancer, not a replacement for the essential 'human spark.' CITAM provides a customized extension of TAM for creative industries, offering practical guidance. The findings can help developers design tools that preserve the agency of the creator and inform policy makers about balancing AIGC innovation with creator rights. These discoveries offer an initial framework for the adoption of ethical AI in the global creative economy, calling for cross-cultural validation to improve generalizability in AI-driven creative ecosystems.

## Introduction

### Research background and problem statement

With a market size exceeding 30 billion in 2023, China's burgeoning micro-short drama sector, a fast-paced, highly interactive context where AIGC integration shows

**Data availability statement:** All data supporting this study are openly available in the Zenodo repository at https://doi.org/10.5281/zenodo.17216841. The dataset includes de-identified survey responses, questionnaire instruments, and interview transcripts with all identifying information removed.

**Funding:** The author(s) received no specific funding for this work.

**Competing interests:** The authors have declared that no competing interests exist.

significant disruptive potential [1,2] -exemplifies the transformative impact of generative AI in global creative industries, but underscores the paradox of autonomy, where efficiency gains clash with control concerns. Breakthroughs in these technologies challenge human-dominated paradigms with tools that automate and enhance content creation [3]. Here, creators use AI for tasks such as script generation to accelerate ideation while retaining personal style, positioning AI as a collaborative partner [4]. This dynamic presents an autonomy paradox: AIGC improves efficiency, but threatens creative control, amplified in China's collectivist culture[5] where group harmony can increase tensions between productivity gains and individual autonomy.

This trend reveals critical theoretical gaps. Although AIGC increases productivity, its implications for the professional identity and autonomy of creators remain underexplored. This creates a tension, framed by debates on technological substitution [6] versus symbiotic collaboration [7]. Ethical issues such as intellectual property and bias further complicate this balance [8]. In the context of micro-short drama, tight production cycles make AIGC appealing, yet the demand for unique narratives necessitates 'conditional adoption', where gains depend on creators retaining control. This challenges the utility-focused assumptions of traditional models like Davis's [9] Technology Acceptance Model (TAM), whose core constructs—Perceived Usefulness (PU) and Perceived Ease of Use (PEOU)—fail to capture the deeper psychological drivers at play in intrinsically motivated creative work. To address this gap, we foreground two more salient factors for this context. Innovation compatibility (IC), alignment of AIGC with creative workflows [10], and Creative Autonomy Retention (CAR), preservation of creative control [11]. Drawing on the literature on human-AI teaming [7], we explore this change from AI as a threat to a partner, emphasizing that control is the key to enhancing creative output. This sets the stage for the 'autonomy paradox': a phenomenon where a high need for creative autonomy, while directly motivating adoption, can paradoxically suppress the positive effects of innovation compatibility when that autonomy feels threatened [10,12].

### Research objectives and theoretical contributions

This study proposes the Creative Industries Technology Acceptance Model (CITAM), which builds upon the foundational ideas of TAM while tailoring the model to the unique psychological and cultural factors relevant to the creative industries. CITAM integrates IC and CAR to explain adoption decisions in creative fields, addressing gaps where traditional models overlook psychological drivers. Introduces IC as the perceived fit of AIGC with creative workflows and CAR as a paradoxical factor, amplifying adoption directly but creating tension in its interaction with other drivers. We also explore how China's cultural context can shape these dynamics, highlighting non-Western uniqueness without overgeneralization.

CITAM distinguishes itself from existing frameworks such as the Unified Theory of Acceptance and Use of Technology 2 (UTAUT2) by uniquely integrating the paradox of autonomy of SDT to fill cultural gaps in creative industries, emphasizing psychological retention over broad environmental factors. Unlike TAM-TRI models that

focus on general trust, CITAM bridges rational adoption drivers with the nuanced tensions of creative autonomy, providing a tailored lens for non-Western contexts where collectivism amplifies paradox effects. This integration advances theoretical positioning by addressing how autonomy needs moderate innovation compatibility, offering a more comprehensive extension than previous TAM variants.

Theoretically, CITAM integrates technology acceptance with creativity psychology to explain AIGC adoption where traditional models fail. It contributes a preliminary understanding of AIGC adoption in a non-Western creative ecosystem, addressing empirical gaps on the autonomy paradox. Our mixed-methods approach provides robust and contextualized insights. Practically, CITAM guides industries toward a human-AI symbiosis that supports a sustainable creative economy.

### Research questions

To guide our study, we address the following.

- RQ1: How does the autonomy paradox affect the adoption of AIGC in creative processes?
- RQ2: How does the autonomy paradox moderate the relationship between innovation compatibility and adoption intentions? (e.g., through negative moderation explaining variance in effects).

## Literature review and theoretical foundations

This chapter reviews literature on technology adoption, innovation diffusion, and creativity-technology relationships to build a foundation for the Creative Industries Technology Acceptance Model (CITAM). Focusing on Innovation Compatibility (IC) and Creative Autonomy Retention (CAR), it analyzes post-2020 research [13,14] and critiques the misalignment of traditional models in creative contexts.

### AIGC in creative industries

Building on Diffusion of Innovations (DOI) [10], IC is the perceived alignment of AIGC with user practices, reducing dissonance and influencing adoption [15]. This is captured by questionnaire items such as IC1 ("AIGC brings revolutionary innovation"). Positive studies show that AIGC improves creativity by sparking ideas [4,7,16], and in China promotes sustainability in micro, small, and medium enterprises (MSMEs) [17]. Cross-culturally, collectivism can improve community compatibility [14], while individualism prioritizes personal boundaries [18].

However, challenges such as job risks and identity crises are significant [19–22]. The literature often lacks theoretical integration, with weak connections between TAM and frameworks like Job Demands-Resources (JD-R) [23] while overlooking creativity-specific concerns [24,25]. Ethical issues, particularly AI bias that erodes autonomy, connect to intrinsic motivation theories [12]. Our CAR construct builds on Self-Determination Theory (SDT) by emphasizing control retention, as illustrated by creators' concerns about "losing control" and fears that "AIGC erodes their agency."

Key gaps remain, especially the need for psychological moderators like CAR to explain adoption paradoxes. In China's short drama industry, high objective compatibility can still encounter subjective resistance [14]. Existing models like TAM-TOE [17] or TAM-TRI [13] often overlook how CAR addresses ethical concerns such as narrative homogenization [26]. The industry's fast pace creates tension between AIGC efficiency and collectivism's emphasis on community creativity. While structuration theory demonstrates AIGC's impact on value chains [27], China's cultural context suggests that collective resilience amplifies the need for CAR, contrasting with Western individualism [28].

### AIGC quality concerns and creative integrity

Recent comprehensive reviews highlight growing concerns about AI's impact on creative industries. Anantrasirichai and Bull [29] systematically categorize AI applications in creative contexts into five domains: content generation, information analysis, content enhancement, information extraction, and data compression. While these applications demonstrate AI's

potential as a "tool for augmenting human creativity," they also reveal significant limitations in "open-ended, unconstrained creative tasks" where AI struggles to compete with human creativity [29].

A critical concern emerging from widespread AIGC adoption is what industry practitioners term "AI slop"—low-quality, homogenized content that lacks artistic depth and originality. Research addresses this phenomenon by arguing that while computers cannot create art in the true sense (as art requires social intention and expression from conscious agents), AI tools represent powerful new instruments for human artists, similar to how photography and animation technologies initially sparked concerns but ultimately expanded artistic possibilities [30].

However, the proliferation of AI-generated content raises quantitative concerns about creative industry sustainability. The rapid production capabilities of AIGC tools can flood markets with low-cost, algorithm-driven content that may undervalue human creativity and compress profit margins for traditional creators. This economic pressure particularly affects fast-paced production environments like China's micro-short drama industry, where efficiency demands may incentivize quantity over quality, potentially leading to the creative homogenization that researchers identify as a key limitation of current AI systems [29].

### Technology adoption models (TAM/DOI)

Although fundamental, the rational choice constructs of the Technology Acceptance Model (TAM): perceived usefulness (PU) and perceived ease of use (PEOU) show diminished relevance in creative contexts where intrinsic factors such as autonomy dominate [9,10]. The extensions of TAM, such as UTAUT and TAM3, maintain a utilitarian logic that often overlooks specific creativity concerns such as identity crises triggered by black-box algorithms [11,27,31–33]. The assumption of linear rationality of TAM fails to capture how creators weigh subjective factors against objective benefits, explaining inconsistent findings in artistic settings. In contrast, CITAM positions itself as an alternative to TAM, specifically designed for creative industries. Unlike TAM and its extensions, CITAM prioritizes psychological factors like Creative Autonomy Retention (CAR) over utilitarian constructs, providing a more nuanced framework for understanding adoption in creative fields. While models like UTAUT2 incorporate broader predictors, they still prioritize performance expectancy over psychological paradoxes, highlighting the novelty of CITAM in focusing on how CAR moderates adoption in creative contexts.

The DOI concept of innovation compatibility [10] often underplays subjective cultural interpretations. In China's collectivist culture, compatibility is socially embedded, putting the harmony of the community above the individualistic limits emphasized in Western contexts [18]. These theories provide a rational foundation, but neglect psychological amplifiers like CAR. Consequently, CITAM is theoretically grounded in the decision to exclude traditional productivity-oriented TAM variables (PU, PEOU). This exclusion is intentional, with the aim of constructing a more parsimonious and context-specific model that prioritizes the core psychological tension between compatibility and autonomy, which we argue is more central to adoption decisions in creative industries than utilitarian calculations. While TAM explains adoption through PU and PEOU, in intrinsically motivated creative fields, these are subsumed by deeper needs like autonomy (SDT), justifying our focus on CAR.

Consequently, CITAM intentionally shifts focus toward psychological mechanisms, particularly SDT's autonomy needs, which are more critical in creative contexts than traditional utility calculations. This theoretical choice is based on the intrinsic motivational characteristics of creative work, reflecting CITAM's context-specific theoretical adaptation.

To highlight these limitations, a summary of key literature gaps includes the following.

1. TAM's overemphasis on rational factors, ignoring creativity's intrinsic drivers (e.g., inconsistent results in artistic domains per [7]);
2. DOI's neglect of subjective paradoxes, such as autonomy threats in high-compatibility scenarios [10];
3. Limited integration of SDT into adoption models, failing to address CAR's moderating role [28];
4. Western bias in studies, overlooking collectivist amplifications of identity tensions [5].

## Autonomy paradox and SDT

The autonomy paradox posits that the retention of creative autonomy (CAR), defined as the perceived ability to maintain control and agency over creative processes [11], exhibits a dual role in the adoption of AIGC. On the one hand, a high need for autonomy can positively drive creators to seek and value AIGC tools that are highly compatible with their workflows [10]. On the other hand, this same high need for autonomy can amplify threat perceptions to creative identity, leading to resistance. Unlike identity threats in previous AI literature [34], our paradox emphasizes the retention of creative autonomy in collective cultures. This tension arises when AIGC interferes with tasks central to the sense of ownership of a professional [35,36]. In scriptwriting, for example, high CAR can transform innovation compatibility (IC) - defined as the alignment of AIGC with creative workflows - from a pure enabler into a potential threat, affecting authorship and motivation [12]. This dynamic integrates the focus of self-determination theory (SDT) on autonomy with the emphasis of diffusion of innovations (DOI) on compatibility, explaining why high CAR can subtly and negatively moderate the IC-adoption relationship when identity fears are evoked [10,28,37].

The paradox manifests itself as a conflict between perceived empowerment and the potential erosion of control. Although AIGC may streamline idea generation, its potential to homogenize outputs can trigger a defensive response, as creators perceive a loss of unique agency. This mechanism is rooted in the emphasis of SDT on intrinsic motivation, where autonomy satisfaction is crucial for engagement, yet AIGC's algorithmic nature can disrupt this by imposing external structures on subjective processes [37]. Evidence from related fields such as AI in education confirms this dual nature, where tools increase efficiency but can erode user agency [13]. Thus, CITAM frames CAR as a psychological amplifier, distinguishing it from other models such as TAM-TRI [13] by stressing its cross-cultural role.

## Cultural contexts in China

China's collectivist culture both intensifies CAR needs and complicates its effects [5]. In collectivist contexts, CAR amplifies the paradox through group-oriented creativity, where threats to shared creative identity are heightened by the importance of group harmony and interdependence, contrasting with Western individual autonomy focus [18]. Here, CAR preserves both individual agency and collective well-being, making adoption more sensitive to relatedness disruptions according to SDT [28].

An editor's insight, 'In our team, AIGC must align with group harmony', illustrates how collectivism intensifies the autonomy paradox by prioritizing communal control. This cultural perspective reveals limitations in Western-centric models, which often overlook how collectivist societies amplify CAR's moderating role, necessitating CITAM's culturally sensitive approach. Future research could employ multigroup analysis to explore these cultural variations empirically.

CITAM addresses these gaps by integrating IC and CAR for culturally sensitive AIGC adoption modeling. Key theoretical distinctions include:

1. TAM/DOI focus on rational compatibility but miss SDT's autonomy mechanisms;
2. SDT emphasizes motivation yet lacks adoption integration;
3. Cultural models highlight collectivism but underexplore paradoxes;
4. CITAM uniquely combines these perspectives, prioritizing CAR's moderation in non-Western creative settings, filling gaps in models like TAM-TOE by emphasizing psychological factors over environmental ones [17].

## Theoretical framework and hypotheses

This study proposes the Creative Industries Technology Acceptance Model (CITAM), which builds upon the foundational ideas of TAM while integrating psychological mechanisms from Self-Determination Theory (SDT) to explain AIGC adoption behavior in creative contexts. CITAM introduces Innovation Compatibility (IC) and Creative Autonomy Retention (CAR) as two core constructs to address the limitations of traditional models in creative industries. The model focuses on

exploring the interaction between compatibility and autonomy in the micro-short drama industry, particularly the autonomy paradox where AIGC enhances efficiency but threatens creative control.

**Core theoretical foundations.** The Technology Acceptance Model (TAM) provides a foundational framework for understanding technology adoption, but reveals limitations in creative contexts. TAM's core constructs—perceived usefulness and perceived ease of use—primarily apply to extrinsically motivated tasks, while creative work is inherently driven by intrinsic motivation. This limitation necessitates integrating Self-Determination Theory to explain technology acceptance behavior among creative professionals.

Self-Determination Theory (SDT) identifies three basic psychological needs that drive human behavior: autonomy, competence, and relatedness [28]. In creative work, the autonomy need is particularly critical, referring to the degree to which individuals perceive their behavior as self-determined and self-guided. SDT distinguishes different types of motivational regulation: intrinsic regulation (behavior is inherently satisfying), identified regulation (behavior aligns with personal values), introjected regulation (acting to avoid guilt or anxiety), and external regulation (acting for rewards or to avoid punishment). Creative work is primarily driven by intrinsic regulation, and when external tools are perceived as "controlling" rather than "informational," a "motivational crowding-out effect" occurs, undermining intrinsic motivation [28].

In the AIGC context, when creators perceive AI tools as threatening their creative autonomy, defensive reactions are triggered even if the tools are functionally powerful. This psychological mechanism forms the theoretical foundation for understanding technology acceptance paradoxes in creative industries and is the fundamental reason why the CAR construct plays a moderating role. CITAM integrates TAM's technology acceptance logic with SDT's autonomy mechanisms to explain complex adoption behaviors in creative contexts.

## Model diagram

Fig 1 illustrates CITAM, showing that IC positively influences adoption intention (AI) and actual usage behavior (AUB), with AI mediating this link. CAR is hypothesized to positively affect IC, but negatively moderate IC's effects on AI and AUB, embodying the autonomy paradox. The arrows indicate proposed relationships, and the moderation is shown as dashed lines. This visual simplifies the model structure for clarity.

This section develops hypotheses based on theoretical foundations, focusing on the relationships among innovation compatibility (IC), adoption intention (AI), actual usage behavior (AUB), and creative autonomy retention (CAR). Each hypothesis is presented with its theoretical basis and a clear statement, grounded in established theories and illustrated with qualitative insights from industry practitioners to contextualize the theoretical mechanisms without revealing empirical findings. Building on DOI, IC mediates the effect of CAR on adoption, since compatibility aligns AI with the autonomy needs of creators, providing a process mechanism where perceived fit translates psychological retention into behavioral results.

**Proposition 1: The autonomy paradox in AIGC adoption.** The autonomy paradox posits that AIGC, while enhancing creative freedom through compatibility, can paradoxically threaten autonomy retention, leading to resistance. This is based on self-determination theory (SDT), where external tools can undermine intrinsic motivation if perceived as controlling [37]. In creative industries, the high need for creative autonomy retention (CAR) amplifies this paradox by increasing identity threats, resulting in weak adoption despite perceived benefits. For example, preliminary interview themes reveal that creators feel 'empowered yet controlled', illustrating how positive attributes of compatibility are offset by underlying fears of autonomy. This core proposition frames the subsequent hypotheses, emphasizing the dual role of CAR as both a potential enabler and a barrier.

**The influence of innovation compatibility.** Drawing on the theory of diffusion of innovations (DOI), innovation compatibility (IC) is a primary determinant of technology adoption [10]. IC refers to the degree to which an innovation is perceived as consistent with the existing values, past experiences, and needs of potential adopters. In the micro-short drama

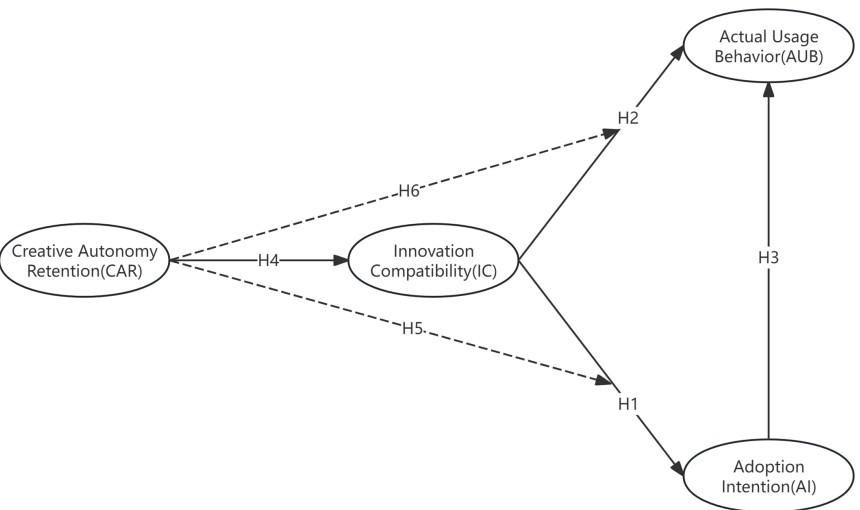

**Fig 1**. **Conceptual model of the creative industries technology acceptance model (CITAM) in the context of AI-generated content (AIGC) adoption in China's micro-short drama industry.** Latent constructs include Innovation Compatibility (IC), Adoption Intention (AI), Actual Usage Behavior (AUB), and Creative Autonomy Retention (CAR). Solid arrows indicate hypothesized direct effects; dashed arrows indicate hypothesized moderating effects. Hypothesis numbers (H1–H6) are displayed alongside the corresponding paths.

industry, a highly compatible AIGC tool would seamlessly integrate into existing creative workflows, such as script generation or visual asset creation, thereby reducing psychological resistance and cognitive friction. This cognitive fit [15] makes the technology feel like a natural extension of the creator's process, fostering positive attitudes. This positive perception is expected to directly shape both the intention to use the technology and its eventual, sustained usage, as a compatible tool minimizes the costs of adaptation and learning. The process mechanism involves compatibility by reducing perceived barriers, thereby facilitating a smoother transition from evaluation to action.

- **Hypothesis 1 (H1):** Innovation compatibility has a significant positive impact on the intention to adopt AIGC.
- **Hypothesis 2 (H2):** The compatibility of innovation has a significant positive impact on the actual usage behavior of AIGC.

  **The mediating role of adoption intention.**  Consistent with the foundational logic of the Technology Acceptance Model (TAM) and the Theory of Reasoned Action, behavioral intention is a robust and immediate predictor of actual behavior [9,38]. In the context of AIGC, the intention formulated by a creator to adopt the technology serves as a critical psychological commitment. This intention bridges the gap between their initial evaluation of the tool (for example, its compatibility) and the ultimate action of incorporating it into their work. As a founder noted in an interview, "strong intention drives consistent behavior", highlighting that a conscious decision to adopt is necessary to overcome inertia and perceived risks. The process mechanism here is the intention that serves as a motivational pivot, converting abstract perceptions into concrete behavioral plans.

- **Hypothesis 3 (H3):** The intention of adoption has a significant positive impact on actual usage behavior of AIGC.

  **The enabling role of creative autonomy retention.**  Based on Self-Determination Theory (SDT) and the locus of control theory, the desire to retain creative autonomy (CAR) is a core psychological need for creative professionals [37,39]. We propose that CAR positively influences the perception of innovation compatibility. When creators feel their agency and

control are secure, they are more likely to view AIGC not as a replacement, but as a supportive tool that can be aligned with their personal creative vision. This sense of control allows them to frame the technology as a collaborator rather than a competitor, thus enhancing its perceived compatibility. The insight of a digital media lecturer supports this: 'Retaining autonomy increases my feelings of compatibility with new tools.' The process mechanism operates through CAR, fostering an open mindset, which in turn amplifies the recognition of alignment benefits.

- **Hypothesis 4 (H4):** The retention of creative autonomy has a significant positive impact on innovation compatibility.

**The paradoxical moderating role of creative autonomy retention.** Here lies the theoretical core of the autonomy paradox. Although CAR can be an enabler (H4), it can also trigger defensive psychological mechanisms. Drawing on research on creative identity [40] and psychological ownership [36], we argue that high CAR needs can amplify threat perceptions. When a creator's sense of authorship and control is paramount, even a highly compatible AIGC tool can be perceived as an intrusion that encroaches upon their creative domain. This creates cognitive dissonance [41], a state of psychological discomfort that the creator seeks to resolve. One way to resolve this dissonance is to downplay the benefits of the technology and resist its adoption, thereby protecting their creative identity. Consequently, the positive influence of compatibility on adoption intention is expected to be attenuated for people with a high CAR. As a CEO insightfully noted, 'high autonomy needs can block AIGC if it feels like a threat'. The process mechanism for this moderation involves increased vigilance, where CAR intensifies the scrutiny of compatibility, weakening its motivational pull.

This defensive mechanism is also expected to impact actual usage. High-CAR individuals may experience increased cognitive demands [42] and identity protection responses [43] when using AIGC, as they must constantly negotiate control boundaries. This added cognitive load and the underlying drive to protect their unique creative identity can lead to passive resistance or reduced engagement, thus weakening the effect of compatibility on actual behavior. Therefore, the very factor that defines a creative professional—their desire for autonomy—paradoxically becomes a barrier to adopting tools designed to augment their creativity. The process mechanism here involves ongoing identity negotiation, which dampens the direct translation of compatibility into sustained use.

- **Hypothesis 5 (H5):** Retention of creative autonomy negatively moderates the effect of innovation compatibility on the intention of adoption, so that the positive effect is weakened when the retention of creative autonomy is high.
- **Hypothesis 6 (H6):** Retention of creative autonomy negatively moderates the effect of innovation compatibility on the actual usage behavior, so that the positive effect is weakened when the retention of creative autonomy is high.

## Methodology

This chapter outlines the methodological framework used to empirically test the Creative Industries Technology Acceptance Model (CITAM) in the context of AIGC adoption in China's micro-short drama sector (MSD). An exploratory sequential mixed methods design [44] was adopted, integrating quantitative structural equation modeling (SEM) with qualitative semi-structured interviews. This approach enables triangulation, enhancing the validity and depth of the findings [45]. The quantitative phase tests the CITAM path and hypotheses, while the qualitative phase explores nuances, such as role-specific adaptations and psychological mechanisms.

### Overall research design

The study follows an exploratory sequential design: (1) qualitative semi-structured interviews to explore key themes and refine the survey instrument; (2) quantitative survey for broad hypothesis testing and model validation. Integration occurs in interpretation, where qualitative insights explain quantitative results (e.g., why CAR moderates IC-AI paths) and inform the general understanding of CITAM constructs. This design addresses common limitations in adoption studies, such

as snapshot data, providing temporal evidence where possible, although the core data remain cross-sectional [46], thus strengthening causal inferences through observed patterns in constructs such as AI and AUB. Triangulation through interviews provides depth to mitigate cross-sectional biases, emphasizing the complementary strengths of mixed methods.

## Quantitative research design

**Sample and sampling procedure.** A priori power analysis (G*Power) confirmed that the sample size of 607 was sufficient to detect small effects in SEM with a power greater than 0.95. A multistage sampling strategy targeted MSD practitioners in China. Stage 1: Purposive selection from industry databases (e.g., WeChat groups, Douyin communities) and platforms like Kuaishou, yielding 607 potential respondents. Stage 2: Attempted stratified sampling by role and experience (novice <2 years, experienced >2 years), though the final sample distribution reflected actual availability and response rates rather than predetermined quotas. Data collection occurred online through Wenjuanxing between 15 May 2025 and 20 June 2025, following the completion of the qualitative interviews. The surveys through Wenjuanxing resulted in 607 complete responses, as the platform requires full submission. Non-response bias was checked using the Armstrong and Overton method, showing no significant differences between early and late respondents [47]. After screening for outliers using the Mahalanobis distance, all 607 responses were considered valid.

Demographics: 42.34% male, 41.19% female, 16.47% unwilling to disclose; average age 33.4 years; creation experience: 30.81% <1 year, 31.63% 1-3 years, 10.87% 4-6 years, 26.69% 7+ years (with 37.56% having 3+ years in MSD); primary roles: 18.62% scriptwriters, 31.8% directors, 18.12% editors, 19.77% independent creators, 11.7% others. The achieved sample distribution differed from initial stratification plans due to varying response rates across professional roles, which may limit the generalizability of findings across all creator types in the micro-short drama industry.

**Measurement instrument.** The measurement instrument focused on CITAM's core constructs (IC, CAR, AI, AUB), with traditional elements of TAM deliberately omitted based on the theoretical rationale for parsimony in this creative context. Items were adapted from established scales and tailored to micro-short drama contexts. Expert reviews and a pretest with practitioners confirmed reliability and validity. Adaptations included translation and cultural adjustments. The complete questionnaire is reproduced in the appendix, with scale development for CAR detailed in Appendix, including item generation from SDT principles and pilot testing.

The key constructs and their psychometric properties are summarized below.

- **Innovation Compatibility (IC):** 4 items adapted from established scales [10,31] (for example, 'AIGC brings revolutionary methods of innovation to the creation of micro-short dramas.' Cronbach's $\alpha$ = 0.901, CR = 0.908, AVE = 0.713.
- **Creative Autonomy Retention (CAR):** 5 original items based on self-determination theory (e.g., 'When using AIGC, I can still maintain full control over core creative decisions'). Cronbach's $\alpha$ = 0.922, CR = 0.922, AVE = 0.702.
- **Adoption Intention (AI):** 4 items from [31] (for example, 'I intend to continue using AIGC in my future micro-short drama projects.' Cronbach's $\alpha$ = 0.887, CR = 0.890, AVE = 0.696.
- **Actual Usage Behavior (AUB):** 3 items from [31] (e.g., 'I use AIGC tools in most of my micro-short drama projects.') Cronbach's $\alpha$ = 0.870, CR = 0.872, AVE = 0.696.

All elements used a 5-point Likert scale (1=strongly disagree to 5=strongly agree). The overall instrument demonstrated excellent reliability (Cronbach's $\alpha$ = 0.901). All constructs exhibited strong convergent validity, with factor loadings exceeding 0.7, composite reliability (CR) values above the 0.7 threshold, and average variance extracted (AVE) values well above the 0.5 benchmark. Discriminant validity was also established (details in Chapter 5). The controls included demographic and environmental factors.

**Quantitative data analysis.** Covariance-based SEM (CB-SEM) was conducted using AMOS 26, which is suitable for confirmatory models, normal data distributions, and larger samples (N=607) [48]. This method was selected for its strength in theory testing and its provision of global fit assessment. The analysis was carried out in two steps: (1) The

measurement model was evaluated using a confirmation factor analysis (CFA) to confirm the reliability and validity of the constructs. (2) The structural model was assessed to test the hypothesized relationships, examining the path coefficients, their significance, and the overall explanatory power of the model ($R^2$). Hypothesis testing used maximum likelihood estimation with bootstrapping procedures for a robust assessment of mediation and moderation effects. As an alternative robustness check, PLS-SEM was run in SmartPLS, yielding consistent results, further validating the findings. Future studies should use IVs (e.g., exogenous AI training such as IV for CAR) to address potential biases.

### Qualitative research design

Before the quantitative phase, semi-structured interviews were conducted to provide exploratory insights, refine the questionnaire, and ensure the validity of the content. This phase aimed to elucidate mechanisms (for example, how CAR influences AI) and validate quantitative findings, with qualitative insights directly forming the interpretation of SEM paths. The small sample (n=10) is acknowledged as a limitation, but thematic saturation was achieved to ensure sufficient depth.

**Participant selection and interview process.** A purposeful sampling of 10 creators with different levels of experience was carried out. Semi-structured interviews (30-45 minutes) were conducted via Tencent Meeting, recorded with consent, and transcribed verbatim for anonymity. The questions included 'How does AIGC affect your creative autonomy?' and 'What factors influence your intention to use AIGC?' Data collection ceased when thematic saturation reached and no new main themes emerged from subsequent interviews. The full interview protocol is in the appendix. n=10 sufficient for thematic depth in the exploratory phase, per Guest et al. [49].

**Qualitative data analysis.** Thematic analysis followed the Braun and Clarke framework [50], which involved familiarization, coding, theme generation, review, and definition. NVivo 12 facilitated the coding process, with inter-coder reliability established (Kappa > 0.8) by two independent coders. The resulting themes were then mapped to the CITAM constructs to triangulate and enrich the quantitative results.

### Ethics approval and consent to participate

This study was reviewed by the Academic Committee of Jinling Institute of Technology (JIT), Nanjing, Jiangsu, China, which determined that the protocol met the criteria for exemption from full ethics review because it involved minimal risk, only adult participants, and anonymized, non-sensitive data. The exemption was granted on 26 February 2025. The committee did not issue a formal approval or exemption reference number.

Qualitative interview data were collected first, between 5 March 2025 and 13 March 2025. Quantitative survey data were subsequently accessed and collected between 15 May 2025 and 20 June 2025 via the Wenjuanxing online platform. The authors did not have access to any direct identifiers of participants during or after data collection; all responses were recorded anonymously and stored in de-identified form.

Informed consent was obtained from all participants prior to data collection. For the interviews, participants signed a written "Interview Informed Consent Form" acknowledging the audio recording and their rights, including voluntary participation and the ability to withdraw without consequence. For the online questionnaire, participants were first presented with an information sheet explaining the objectives, procedures, voluntary nature of participation, data confidentiality, and the right to withdraw at any time without penalty. They could proceed to the questionnaire only after clicking an "I agree to participate" button.

No minors (under 18 years of age) participated in this study.

### Robustness checks

The robustness of the findings was verified through multiple tests. Harman's single factor test indicated that common method bias (CMB) was not a significant concern, as the first factor explained less than 50% of total variance [51]. In addition to the commonly used Harman's single-factor test, this study also employed latent method factor analysis.

This method adds a common method factor to the measurement model to assess the impact of method bias on construct loadings and compares the fit improvement with and without the method factor. The latent method factor analysis showed that the method factor explained an average variance of 18.6% (below the 25% threshold), with minimal model fit improvement ($\Delta$CFI = 0.009, $\Delta$RMSEA = –0.011), and the average correlation coefficient among variables was 0.376 (<0.50 threshold). To mitigate CMB, we used procedural remedies and statistical tests; Interviews (n = 10) ensured thematic saturation. Multicollinearity was assessed and all the Variance Inflation Factor (VIF) values fell well below the 5-threshold, supporting the integrity of the model. Further checks included addressing potential self-report biases through assurances of anonymity and a careful questionnaire design. Alternative model specifications were also tested to confirm the parsimony and superiority of the proposed CITAM framework. These measures collectively mitigate potential biases and reinforce the validity and generalizability of the study conclusions. However, the predominantly cross-sectional design remains a limitation, and future research could incorporate objective measures (e.g., usage logs) to complement self-reports for stronger causal evidence and to address self-report deviations. Subgroup analyses (e.g., by gender and role) confirmed model stability, despite the acknowledged sample biases.

## Data analysis and research findings

This chapter presents the empirical findings of our exploratory sequential mixed-methods design. We first report the quantitative results of the structural equation modeling (SEM) analysis, which tests the hypotheses (H1–H6) of the Creative Industries Technology Acceptance Model (CITAM). Following this, we present the thematic analysis of our qualitative interviews, which provides rich and contextual insights into the adoption dynamics. Finally, we integrate both data sets to offer a comprehensive understanding of the adoption of AIGC in creative industries, with a focus on Innovation Compatibility (IC) and Creative Autonomy Retention (CAR).

### Sample characteristics and data collection

Data was collected from 607 Chinese creative professionals, including scriptwriters, directors, editors, independent creators, and others. The sample was diverse: 42.34% male, 41.19% female, and 16.47% unwilling to disclose, with an average age of 33.4 years (SD = 5.8). Experience levels were 30.81% with <1 year, 31.63% with 1-3 years, 10.87% with 4-6 years, and 26.69% with 7+ years. Roles included 18.62% scriptwriters, 31.8% directors, 18.12% editors, 19.77% independent creators, and 11.7% others.

The stratified sampling plan aimed for 40% scriptwriters, 30% directors, 20% producers, and 10% others. The final distribution showed more directors (31.8%) and independent creators (19.77%) than planned, with fewer scriptwriters (18.62%) and editors (18.12%). Producers were not measured, which may affect sample representativeness. These discrepancies should be considered when interpreting results. The analysis was performed using SPSS 26.0 and AMOS 26.

**Measurement model validity.** The measurement model showed strong validity. Convergent validity was confirmed with Average Variance Extracted (AVE) scores all exceeding 0.696 and Composite Reliability (CR) scores all above 0.872. Discriminant validity was established using the Fornell-Larcker criterion and by ensuring that all heterotrait-monotrait (HTMT) ratios were below the 0.85 threshold [52]. The confirmatory factor analysis (CFA) demonstrated an excellent fit to the model: $\chi^2$ = 179.729, df = 98, $\chi^2$ / df = 1.834, CFI = 0.988, TLI = 0.985, RMSEA = 0.037 and SRMR = 0.030 [53]. Additional metrics of the validity analysis include factor loadings ranging from 0.761 to 0.923, a KMO of 0.900, and a significant Bartlett's test (p < 0.001), confirming robust measurement properties.

Discriminant validity was established using the Fornell-Larcker criterion, where all inter-construct correlations were lower than the square root of each construct's AVE (see Appendix). Heterotrait-monotrait (HTMT) ratios were all below the 0.85 threshold, further confirming discriminant validity. Descriptive statistics revealed construct means ranging from

3.24 to 3.78 (on a 5-point scale), with moderate standard deviations (0.84-1.18), indicating normal data distribution without extreme outliers. Complete factor loadings, discriminant validity matrix, and descriptive statistics are provided in the appendix.

**Descriptive statistics and construct correlations.** Descriptive statistics and inter-construct correlations are presented in Table 1. All construct means ranged from 3.48 to 3.72 (on a 5-point scale), indicating moderate to high levels across all variables. Standard deviations were moderate (0.95-0.99), suggesting adequate variability. Discriminant validity was established using the Fornell-Larcker criterion, where all inter-construct correlations were lower than the square root of each construct's AVE. The highest correlation was between CAR and IC (r = 0.424), which remained below the threshold, confirming adequate discriminant validity.

**Structural model assessment.** The structural model also showed an excellent fit ($\chi^2$ / df = 2.413, CFI = 0.980, TLI = 0.976, RMSEA = 0.048, SRMR = 0.035), indicating that the theoretical model aligns well with the empirical data. Fig 2 illustrates the path coefficients, while Table 2 summarizes the hypothesis testing results.

All direct effect hypotheses were supported. IC was positively associated with AI (H1: $\beta$ = 0.199, p < 0.001, $f^2$ = 0.04 indicating a small effect) and AUB (H2: $\beta$ = 0.235, p < 0.001, $f^2$ = 0.06 indicating a small effect). The intention-behavior link was validated (H3: AI → AUB, $\beta$ = 0.299, p < 0.001, $f^2$ = 0.09 indicating a small-to-medium effect). The path from CAR to IC (H4: $\beta$ = 0.456, p < 0.001, $f^2$ = 0.26 indicating a strong medium-to-large effect) emerged as the most powerful predictor in the model, suggesting that securing the autonomy of the creator is a critical prerequisite for them to perceive AIGC as compatible.

The moderation hypotheses, central to the autonomy paradox, were also supported. CAR negatively moderated the IC-AI relationship (H5: $\beta$ = –0.076, p < 0.05, $f^2$ = 0.006) and the IC-AUB relationship (H6: $\beta$ = –0.078, p < 0.05, $f^2$ = 0.006). Although $f^2$ = 0.006 indicates a small effect, it remains theoretically significant, aligning with the modest $\Delta R^2$ increase and reflecting a subtle influence in the moderation process.

For moderation analysis, to mitigate multicollinearity concerns, CAR and IC variables were mean-centered prior to constructing interaction terms (CAR_centered = CAR - M_CAR; IC_centered = IC - M_IC). The interaction term was constructed as the product of centered variables (CAR_centered × IC_centered). Post-centering VIF values were all below 2.5, confirming that multicollinearity was not a major concern. Maximum likelihood estimation employed bootstrapping procedures for robust assessment of mediation and moderation effects. Power analysis suggests a larger sample size for replication. Although modest, these effects highlight subtle paradoxes in creative adoption, with statistical significance (with CIs not including zero) revealing a crucial dynamic.

The model explained significant variance in endogenous variables: $R^2$ = 0.208 for IC, $R^2$ = 0.190 for AI, and an impressive $R^2$ = 0.335 for AUB. The substantial variance explained in actual usage highlights the strong explanatory power of the model. Although the effect size is small ($f^2 \approx 0.006$), the interaction term (CAR × IC) explains a small but statistically significant increase in variance, with a $\Delta R^2$ of 0.006. This suggests that the moderation effect, while modest, contributes to the variance in the moderated relationships, as shown in Fig 2.

This small but significant moderation underscores the nuance of the paradox in the real world: In creative industries, where decisions are influenced by intricate identity and control factors, even minor attenuations can accumulate to shape

**Table 1**. Descriptive statistics, correlations, and discriminant validity assessment.

| Construct | M | SD | IC | CAR | AI | AUB |
|-----------|------|------|-------|-------|-------|-------|
| IC | 3.72 | 0.95 | 0.845 | | | |
| CAR | 3.48 | 0.97 | 0.424 | 0.838 | | |
| AI | 3.71 | 0.96 | 0.290 | 0.337 | 0.845 | |
| AUB | 3.71 | 0.99 | 0.333 | 0.358 | 0.320 | 0.834 |

*Note: Diagonal elements are square roots of AVE; off-diagonal elements are inter-construct correlations; all correlations significant at p < 0.01.*

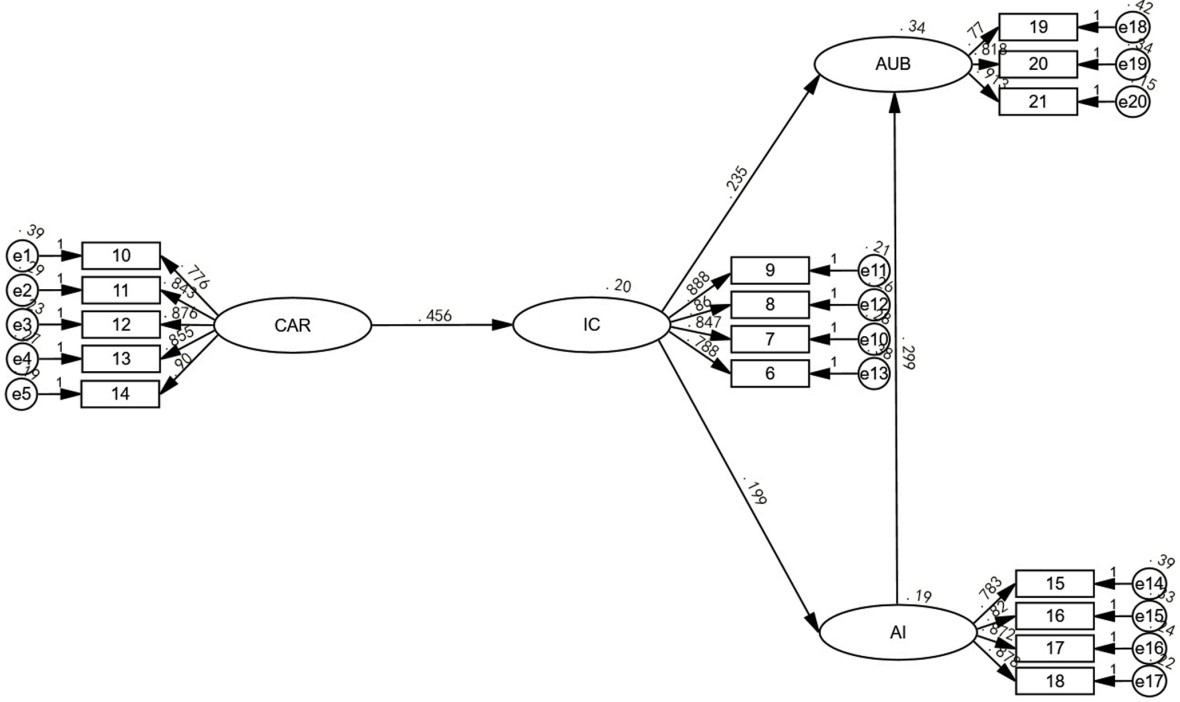

**Fig 2. Structural equation modeling (SEM) results for the Creative Industries Technology Acceptance Model (CITAM).** Latent variables— Innovation Compatibility (IC), Adoption Intention (AI), Actual Usage Behavior (AUB), and Creative Autonomy Retention (CAR)—are measured by multiple observed indicators (items 6–21) with associated error terms (e1–e21). Standardized path coefficients are shown for all relationships between latent and observed variables, as well as between the latent constructs themselves. Solid lines represent statistically significant paths; dashed lines represent non-significant paths. All coefficients are standardized estimates. Model fit indices and significance levels are reported in the Results section.

**Table 2. Hypothesis testing results.**

| H | Path | β | z/t-value | p-value | $f^2$ | 95% CI | Sup. |
|---|---|---|---|---|---|---|---|
| H1 | IC → AI | 0.199 | 4.185 (z) | <0.001 | 0.040 | [0.10, 0.29] | Yes |
| H2 | IC → AUB | 0.235 | 5.134 (z) | <0.001 | 0.059 | [0.14, 0.33] | Yes |
| H3 | AI → AUB | 0.299 | 6.623 (z) | <0.001 | 0.091 | [0.20, 0.40] | Yes |
| H4 | CAR → IC | 0.456 | 10.176 (z) | <0.001 | 0.260 | [0.36, 0.55] | Yes |
| H5 | CAR mod. IC → AI | −0.076 | −2.026 (t) | <0.05 | 0.006 | [−0.15, −0.00] | Yes |
| H6 | CAR mod. IC → AUB | −0.078 | −2.155 (t) | <0.05 | 0.006 | [−0.16, −0.00] | Yes |

*Note: All hypotheses supported.*

overall adoption patterns. For example, the negative values of β (−0.076 and −0.078) indicate a persistent resistance undercurrent that, while not overpowering, systematically weakens the benefits of compatibility for high-CAR creators. This transforms the 'small' size into a theoretical strength, revealing how paradoxes operate as quiet barriers rather than dramatic ones, aligning with the overall model $R^2$ for AUB (0.335) and providing a foundation for future refinements in similar contexts. Such subtlety is particularly relevant in high-stakes creative work, where psychological factors like autonomy can subtly erode enthusiasm without fully stopping adoption, thus explaining unmodeled variance in traditional models.

**Moderation and mediation analyses.** To further probe the autonomy paradox, we conducted a simple slope analysis. For H5, the analysis revealed that for creators with low CAR (–1 SD), the positive effect of IC on AI was strong and significant (β = 0.235, 95% CI [0.162, 0.308], p < 0.001). For those with high CAR (+1 SD), this effect was weakened to

non-significance ($\beta$ = 0.094, 95% CI [–0.007, 0.195], p = 0.068). However, for those with high CAR, this effect was weakened to the point of non-significance ($\beta$ = 0.094, p = 0.068). This pattern provides compelling evidence for the paradox: the very individuals who value autonomy the most are so sensitive to its potential erosion that the positive influence of a tool compatibility on their adoption intention is effectively nullified. For H6, a similar pattern emerged: at low CAR, IC strongly predicted AUB ($\beta$ = 0.312, p < 0.001), but at high CAR, the effect diminished significantly ($\beta$ = 0.156, p < 0.05). These slopes illustrate how high autonomy concerns can override compatibility benefits, leading to lower usage behavior. The analysis used non-standardized coefficients for clarity, with z-scores confirming significance (e.g., for IC→AI at low CAR: z=8.144, p = 0).

Fig 3 illustrates the simple slope plots for the modulating effects of CAR on the IC-AI and IC-AUB relationships, visually depicting how much CAR dampens the positive slopes. The shaded areas represent 95% confidence intervals, highlighting the non-overlap at high CAR levels, which underscores the impact of the paradox.

Bootstrapping confirmed the stability of all paths. The mediation analysisusing 5,000 bootstrap samplesconfirmed a significant indirect effect for the path CAR → IC → AUB ($\beta$ = 0.107, 95% BootCI [0.068, 0.151]), underscoring the importance of CAR in shaping behavior through its influence on compatibility perceptions. The model explains 33. 5% of the AUB variance, with CAR showing a significant indirect effect via IC ($\beta$ = 0.107, p < 0.01).

**Robustness checks.** To further examine common method bias, multiple methods were used. Harman's single-factor test showed the first factor explained 23.4% of the variance (<50% threshold) [51]. Latent method factor analysis revealed an average method factor variance of 18.6% (below 25%), with slight improvements in model fit indices (ΔCFI = 0.009, ΔRMSEA = –0.011). The average inter-variable correlation was 0.376 (<0.50 threshold). Procedural controls, including anonymous questionnaires and reverse-coded items, were also applied. Evidence suggests common method bias is not a significant concern. Multicollinearity tests showed VIF values below 2.5. The model was replicated using PLS-SEM, yielding consistent results, confirming the stability of CB-SEM findings.

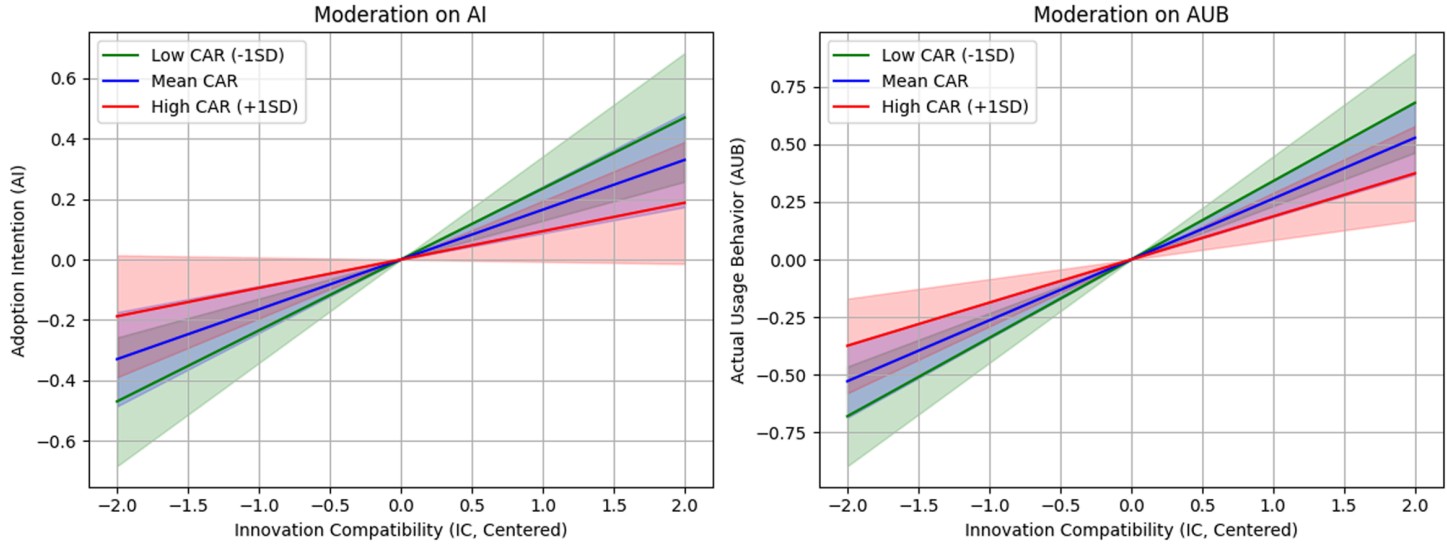

**Fig 3**. **Simple slope analyses illustrating the moderating effect of Creative Autonomy Retention (CAR) on the relationship between Innovation Compatibility (IC) and (A) Adoption Intention (AI), and (B) Actual Usage Behavior (AUB).** High and low levels of CAR are plotted at one standard deviation above (+1SD) and below (–1SD) the mean. Shaded areas represent 95% confidence intervals around the regression lines.

## Qualitative analysis: Thematic insights

Semi-structured interviews with 10 creators were thematically analyzed [50], yielding a Kappa of 0.85 for intercoder reliability. Three primary themes emerged: (1) Creativity Amplification, where AIGC boosts ideas but needs human guidance; (2) Autonomy Concerns, where a high need for control leads to skepticism and fears of losing creative ownership; and (3) Integration Challenges, where shifts in workflow create new barriers related to identity threats and cognitive load.

## Integration of quantitative and qualitative findings

Table 3 summarizes the triangulation of our key findings, juxtaposing quantitative evidence with illustrative qualitative insights.We triangulated the SEM results with the qualitative themes to achieve a holistic understanding. The integration reveals high convergence. For example, the quantitative finding of the negative moderation of CAR (H5, H6) is vividly explained by the qualitative theme 'Autonomy Concerns'. Simple slope analysis, showing the effect of compatibility disappearing for high-CAR individuals, is perfectly mirrored by the director's statement: 'If I feel AI is taking over, I don't care how well it fits my workflow - I will not use it.'

Regarding the theme "Creativity Amplification", qualitative quotes such as 'AI accelerates ideas, but threatens originality' directly explain H5's negative moderation (path coefficient –0.076), illustrating how perceived gains in efficiency are undermined by fears of autonomy, weakening the IC-AI link. In the theme 'Autonomy Concerns', sentiments such as 'High autonomy needs make me hesitate, even if it's a good tool' reinforce the effect of H6 (path coefficient –0.078), showing the extension of the paradox to behavior. For "Integration Challenges", quotes like 'I could try it, but I'll not rely on it if it threatens my role' (Theme 3) highlight workflow barriers that align with the overall model $R^2$, capturing unmodeled nuances in adoption resistance.

Although convergence was high, qualitative data also pointed to unmodeled nuances, such as cultural influences, which could explain some of the remaining variance. In general, the integrated findings position CAR as a critical double-edged element for sustainable AIGC integration.

# Discussion

This chapter interprets the empirical findings by integrating quantitative results with qualitative insights, exploring theoretical contributions, practical implications, and limitations.

## Theoretical mechanisms explained

The autonomy paradox mechanism identified in this study can be explained through the following psychological processes:

**Table 3**. Integration of findings.

| Hypothesis/Path | Quant. Evidence ($\beta$, p, f²) | Qual. Evidence (Theme/Quote) | Interpretation |
| --- | --- | --- | --- |
| H4: CAR → IC | 0.456, <0.001, 0.26 | "When I feel in charge, the tool feels more cooperative." (Theme 2) | Autonomy is a key prerequisite for perceiving compatibility. |
| H5: CAR mod. IC→AI | –0.076, <0.05, 0.006 | "High autonomy needs make me hesitant, even if it's a good tool." | Paradox confirmed: a high need for autonomy weakens the IC-AI link. |
| H6: CAR mod. IC→AUB | –0.078, <0.05, 0.006 | "I might try it, but I won't rely on it if it threatens my role." (Theme 3) | The paradox extends to behavior, dampening actual usage. |
| Model Power | R²(AUB) = 0.335 | "It's a mix of efficiency gains and control fears." (All Themes) | The model captures a third of the variance in usage, validating its core logic. |

*Note: All p < 0.05, control variables standardized.*

First, cognitive evaluation process: Creators initially assess whether AIGC threatens their professional identity and creative control. This assessment is based on SDT's autonomy needs—when tools are perceived as supporting autonomy, creators are more accepting; when perceived as controlling, resistance is triggered.

Second, motivational regulation shift: When threats are perceived, creators' motivational regulation shifts from intrinsic to introjected regulation, generating anxiety and resistance. This explains why even objectively compatible tools may encounter resistance.

Third, behavioral adaptation strategies: High-CAR individuals adopt "limited engagement" strategies, utilizing tool advantages while striving to maintain control. This strategic usage pattern leads to the weakening of compatibility's influence on usage behavior.

This mechanism reveals the limitations of traditional technology acceptance models in creative industries: they ignore the moderating role of professional identity on technology acceptance. The small effect sizes ($f^2 \approx 0.006$) indicate that the paradox operates as a subtle but persistent undercurrent of resistance rather than direct rejection, reflecting the complexity and subtlety of psychological factors in creative decision-making.

## Integration of core findings and theoretical dialogue

Using an exploratory sequential mixed-methods design, this study combines quantitative path analysis with qualitative themes to explain adoption mechanisms among Chinese micro-short drama creators. This integration not only confirms statistical relationships, but also enriches them with real-world narratives, illustrating how abstract constructs manifest themselves in daily creative practices. The discussion focuses on two key insights: the revalidation of core adoption logics in creative contexts and the profound impact of the autonomy paradox.

First, the study confirmed that innovation compatibility (IC) significantly and positively predicted both adoption intention (AI) and actual usage behavior (AUB), supporting H1 and H2. This aligns with foundational theories such as TAM [9] and DOI [10]. The link between intention and behavior (H3) was also validated. Qualitative insights add crucial depth: creators perceive a compatible AIGC not just as a tool, but as an 'inspiration catalyst' that seamlessly integrates and enhances their creative workflows. This extends traditional models by demonstrating that in creative fields, compatibility is deeply intertwined with value harmony and the reduction of creative friction. CITAM addresses TAM limitations by achieving substantially higher explanatory power in creative contexts, addressing gaps where standard models overlook psychological factors in intrinsically motivated domains.

Second, and central to our model, Creative Autonomy Retention (CAR) emerged as a powerful, dual-edged construct, revealing the autonomy paradox. The findings strongly supported H4, showing that CAR showed a substantial positive relationship with IC ($\beta = 0.456$, $f^2 = 0.26$), positioning it as the most critical prerequisite for adoption. Creators must first feel their autonomy is secure before they can even perceive an AI tool as compatible. Simultaneously, the results supported H5 and H6, demonstrating that CAR enacts a subtle, yet statistically significant, negative moderating effect on the relationships between IC and AI and AUB. In low-CAR scenarios, compatibility easily drives adoption. In high-CAR scenarios, this link is weakened by skepticism and identity threat. This extends JD-R theory by framing CAR as a resource-demand tension [23].

This paradox, evident in negative moderations combined with interview quotes like 'I resist if it threatens my autonomy', challenges simplistic human-versus-AI binaries and extends structuration theory [27]. The small effect sizes ($f^2 \approx 0.006$) for these moderations suggest that the paradox operates as a quiet but persistent undercurrent of resistance rather than an outright veto. CITAM advances paradox theory [54] by quantifying tensions as moderators, unlike qualitative foci. The strong explanatory power of the model for actual usage ($R^2 = 0.335$) validates this core logic, showing that this paradoxical tension is a key determinant of real-world behavior.

## AIGC adoption as quality control mechanism

Our findings provide important insights into how the autonomy paradox may serve as a natural quality control mechanism against AI-generated "slop." The negative moderation effects of CAR (H5: $\beta$=-0.076; H6: $\beta$=-0.078) suggest that creators with strong autonomy needs naturally resist over-reliance on AI tools, potentially protecting against the content degradation risks identified by researchers [29]. This resistance may be quantitatively beneficial for industry health, as it preserves human creative input necessary for meaningful artistic expression.

The qualitative theme of "Creativity Amplification" aligns with the perspective that AI tools should augment rather than replace human creativity [30]. Our interview data reveals that successful creators maintain what researchers describe as essential "human agency and intention" in the creative process [30], viewing AIGC as an "inspiration catalyst" rather than an autonomous creator. This approach helps preserve what is identified as art's fundamental requirement for social intentionality and conscious expression [30].

Quantitatively, our model's strong explanatory power for actual usage behavior ($R^2$ = 0.335) suggests that the autonomy paradox operates as a quality filter. High-CAR creators who show reduced AIGC reliance (per the negative moderation effects) may naturally produce more distinctive, human-guided content, addressing concerns about creative homogenization. This selective engagement pattern could help micro-short drama creators maintain competitive advantage through preserved artistic authenticity while leveraging AI efficiency gains.

The economic implications are significant. By maintaining selective rather than wholesale AIGC adoption, creators may avoid the "race to the bottom" in content quality that threatens creative industries. Our findings suggest that psychological factors like CAR serve as natural barriers to the kind of uncritical AI adoption that leads to market flooding with low-quality content.

## Theoretical contributions

This study advances technology adoption theory by introducing the Creative Industries Technology Acceptance Model (CITAM). CITAM builds upon insights from TAM and UTAUT while integrating psychological mechanisms from Self-Determination Theory (SDT) and Diffusion of Innovations (DOI). Specifically, it operationalizes and validates the autonomy paradox by positioning CAR as a dual-role construct: a powerful antecedent to compatibility and a subtle negative moderator of its effects.

This contribution is significant because it quantifies a paradox previously discussed mostly in qualitative terms. Provides empirical evidence for how identity threats and cognitive load [34,42] manifest in adoption models within high-autonomy fields. Table 4 compares CITAM with foundational models, highlighting its gradual extension through focused psychological integration.

CITAM increases the explanatory power over TAM (from our $R^2$ comparison), which aids AI ethics debates. By blending SDT with adoption frameworks, CITAM offers a new lens for fusion of AI-human creativity, validated in China's MSD industry. It contributes to paradox theory [54] by demonstrating how a core professional value, autonomy, can simultaneously enable and hinder innovation.

**Generalization of CITAM.** Beyond creative industries, CITAM holds potential for generalization to noncreative knowledge work domains, such as programming or data analysis, where professionals face similar tensions between AI efficiency and personal control. In these fields, the autonomy paradox may amplify through JD-R frameworks, where CAR acts as a demand that heightens resource strain, potentially weakening adoption paths even more than in creative contexts. For instance, programmers might experience heightened skepticism toward AI code generators if they perceive threats to their problem-solving autonomy, mirroring the negative moderation effects observed here. This future assumption suggests that CITAM could extend to broader knowledge economies by testing CAR's dual role in tasks requiring intellectual ownership, thus amplifying its theoretical reach. Future research might hypothesize that in such domains, the paradox manifests itself with stronger negative effects due to the emphasis on individual expertise, offering a pathway

**Table 4**. Comparison of CITAM with TAM and UTAUT.

| Model | Core Constructs | Key Focus | Extension in CITAM |
|---|---|---|---|
| TAM (Davis, 1989) | Perceived Usefulness, Perceived Ease of Use | Individual technology acceptance in general settings | Adds CAR as a dual-role construct, quantifying the autonomy paradox in creative industries; integrates qualitative insights for context-specific depth. |
| UTAUT (Venkatesh et al., 2003) | Performance Expectancy, Effort Expectancy, etc. | Unified view of acceptance under organizational moderation | Incorporates IC and CAR to address psychological resources (autonomy) and paradoxes lacking in UTAUT; validated in non-organizational, creative contexts with strong explanatory power. |
| CITAM (This Study) | IC, CAR, AI, AUB | Psychological drivers in creative AIGC adoption | Builds upon both, quantifying CAR's paradoxical effect, providing a validated model for innovation-driven fields with strong explanatory power for behavior ($R^2 = 0.335$ for AUB). |

Note: IC = Innovation Compatibility, CAR = Creative Autonomy Retention, AI = Adoption Intention, AUB = Actual Usage Behavior.

to refine CITAM for universal application in AI-human collaboration scenarios. This extension would respond to calls for models that transcend industry silos, improving the impact of CITAM on the literature on technology adoption.

## Practical implications

**Design guidance for AIGC developers.** Drawing on the framework for AI applications in creative contexts, developers should prioritize AI applications that enhance rather than replace human creativity [29]. Specific recommendations include: (1) Implement "autonomy preservation features" such as granular control settings and human-override capabilities that support high-CAR creators; (2) Design content generation tools that require meaningful human input and decision-making, preventing the "AI slop" phenomenon; (3) Develop quality indicators and originality metrics that help creators assess when AI assistance maintains versus compromises artistic integrity. Prioritize features that improve user control, such as 'autonomy sliders' or editable output, to directly support CAR and mitigate resistance. Ethical and transparent design is crucial to build trust and avoid homogenization risks [55].

Following the emphasis on human intentionality in art, AIGC tools should be positioned as "creative instruments" rather than autonomous creators [30]. This requires transparent algorithmic processes that allow creators to understand and guide AI outputs, preserving the human agency that researchers identify as essential for meaningful artistic expression [30].

**Organizational change for creative managers.** Managers should implement what researchers term "human-centric AI integration," emphasizing selective adoption strategies that maintain creative quality [29]. Our CAR findings suggest establishing "autonomy benchmarks" where projects maintain minimum levels of human creative control. This approach addresses both the efficiency demands of micro-short drama production and the quality concerns raised by industry practitioners regarding AI-generated content proliferation. Frame AIGC as a creative partner, not a replacement. Foster symbiosis through customization training and incentives that reward unique, human-guided AIGC applications [56].

**Strategic advice for educators and policymakers.** Policymakers should consider the quantitative impact of AIGC on creative labor markets. Our findings suggest that supporting high-CAR creators through autonomy-preserving policies may naturally regulate against market oversaturation with low-quality AI content. This includes funding programs that incentivize human-AI collaboration models rather than AI replacement strategies, and establishing quality standards that

preserve the human creative elements that researchers argue are essential for authentic artistic expression [30]. Educators must integrate AIGC literacy, ethics, and control retention techniques into curricula. Policymakers should balance innovation with creator rights, mandating transparency in line with global standards like the European Union's Artificial Intelligence Act (EU AI Act, 2024, Articles 13-14) [57], while considering cultural contexts. Our findings inform policy: mandate AIGC transparency to preserve creator autonomy. These suggestions comply with the EU AI Act's transparency requirements, enhancing international relevance. In collectivistic settings like China, the emphasis on AIGC's communal benefits can align with cultural values and lower barriers related to autonomy.

### Limitations and future directions

**Theoretical limitations.** The dimensions of CAR could be further refined (e.g. emotional vs. operational). The small moderation effects, while significant, indicate that the paradox is a subtle mechanism and that other factors are at play, highlighting the need for further theoretical development. Small effects are related to sample power (0.75).

**Methodological limitations.** The cross-sectional data hinder strong causal claims. The China-centric sample limits the generalizability of cultures. Reliance on self-reported measures may introduce response biases, although robustness checks provided some mitigation. The absence of a formal analysis of instrumental variables (IV) for endogeneity remains a limitation.

**Future extensions.** Longitudinal designs are needed to track the evolution of these dynamics; future may adopt longitudinal designs for stronger insights. Cross-cultural research is essential, particularly comparing collectivistic contexts with individualistic Western cultures (e.g., the U.S. film industry), where the autonomy paradox may manifest differently. In individualistic cultures, the paradox may differ, potentially amplifying negative moderations due to a stronger emphasis on personal authorship. Future studies should incorporate objective usage data (e.g., app logs) to overcome self-report limitations and employ experimental designs to establish causality, including innovative AI experiments for deeper insights.

### The role of context and boundary conditions

The applicability of CITAM is bounded by context. The autonomy paradox may be amplified in tasks with high psychological ownership (e.g., solo scriptwriting) and mitigated in more collaborative, low-ownership tasks. The cultural context is paramount: the negative moderating effect of CAR might be stronger in individualistic cultures that value unique authorship. Technological maturity also matters; the paradox may be more pronounced with nascent, less understood AIGC tools. Future research should test these boundary conditions to refine CITAM into a more robust and globally applicable model for understanding human-AI creative collaboration.

## Conclusion

This study, employing a rigorous mixed-method design, constructs and provides initial validation for the Creative Industries Technology Acceptance Model (CITAM) to address a pivotal research question. What are the psychological decision-making mechanisms that underlie the adoption of AIGC by creative professionals in China's dynamic micro-short drama industry? Blending quantitative path analysis with qualitative insights from 607 participants, the research elucidates these mechanisms, confirming the six hypotheses and highlighting Innovation Compatibility (IC) as a driver of adoption, moderated by the paradoxical role of Creative Autonomy Retention (CAR), enhancing compatibility perceptions while simultaneously introducing skepticism. Returning to the research question, the findings reveal how CAR shapes creative decision making amid rapid innovation, providing a pathway to reframe AIGC from a threat to an ally.

The study's implications extend beyond technology adoption to fundamental questions about creative integrity in the AI era. By revealing how the autonomy paradox operates as a quality control mechanism, CITAM offers a framework for understanding how creative industries might naturally self-regulate against the "AI slop" phenomenon identified by industry practitioners. This research demonstrates that psychological factors like CAR may serve as essential barriers to the

content homogenization risks that ANANTRASIRICHAI and BULL [29] associate with uncritical AI adoption in creative contexts.

Following HERTZMANN's [30] argument that meaningful art requires human intention and social expression, our findings suggest that creators who maintain strong autonomy needs naturally preserve these essential human elements even when adopting AI tools. This selective engagement pattern may be crucial for maintaining creative industry sustainability and preventing the economic devaluation that can result from flooding markets with algorithmically-generated content.

Theoretically, CITAM advances beyond TAM by embedding CAR as a dual-role construct, bridging information systems, creativity studies, and organizational behavior. Empirically, the mixed methods approach offers robust evidence, with the model demonstrating moderate explanatory power for adoption behaviors and successfully identifying small but significant paradoxical effects. Practically, the findings inform empowered designs for developers, symbiotic cultures for managers, and balanced ecosystems for policymakers. This emphasizes the key industry impact of the study: providing a clear framework to balance AIGC innovation with creator rights in China to promote sustainable growth.

This study concludes with a forward-looking call for targeted policy actions that mandate AIGC transparency and incentivize user control features to address anxieties around creative ownership. Policymakers are urged to balance AIGC innovation with creator rights in China, promoting regulations that safeguard psychological ownership while advancing technology. In essence, this research illuminates the nuanced fusion of human and artificial creativity, providing essential first steps toward building harmonious, ethical, and sustainable creative ecosystems worldwide. Future replications in other creative sectors are strongly encouraged.

CITAM not only provides a roadmap for China's micro-short drama industry, but also paves the way for ethical AI in the global creative economy, calling for cross-industry validation. By revealing how the autonomy paradox shapes adoption decisions, this study emphasizes the need to preserve human creativity in an AI-dominated era. It offers practitioners initial steps to design autonomy-supporting tools and provides policymakers with insights to cultivate fair innovation environments. Ultimately, this work fosters broader discussions on culturally sensitive AI frameworks, driving a shift from confrontation to collaboration, and laying the foundation for sustainable creative practices.

## Supporting information

**S1 File. Complete questionnaire instrument.** Contains the complete bilingual (Chinese-English) questionnaire used in the quantitative study, including all items for Innovation Compatibility (IC), Creative Autonomy Retention (CAR), Adoption Intention (AI), and Actual Usage Behavior (AUB), along with demographic questions.
(DOCX)

**S2 File. CITAM scale development documentation.** Comprehensive documentation of the scale development process for all CITAM constructs, with detailed focus on the Creative Autonomy Retention (CAR) scale. Includes theoretical grounding, item generation, pilot testing, expert reviews, and psychometric validation.
(DOCX)

**S3 File. Interview outline and transcripts.** Contains the interview outline (question guide) and complete de-identified transcripts from 10 creative professionals. All potentially identifying information has been removed to protect participant privacy.
(DOCX)

**S4 File. Survey dataset.** De-identified survey responses from 607 micro-short drama creators. Dataset includes all questionnaire responses with personal identifiers removed.
(CSV)

**S5 File. Complete statistical analysis results.** Comprehensive statistical output including SEM path coefficients, model fit indices, reliability analyses, validity assessments, moderation analyses, and mediation testing.
(XLSX)

**S6 File. Detailed measurement statistics.** Factor loadings, discriminant validity matrices, descriptive statistics, and psychometric properties for all constructs.
(XLSX)

**S7 File. Study codebook.** Codebook for the CITAM dataset (N = 607, 21 variables), listing variable names, labels, coding, and notes. Includes demographics, IC, CAR (CAR4 reverse-coded), AI, and AUB items. Documents missing value treatment (none), data transformation, and scale composition, with all items measured on 5-point Likert scales.
(DOCX)

## Acknowledgments

We extend our heartfelt appreciation to the editorial and review team whose expertise significantly enhanced the quality of this manuscript. We are particularly grateful to Editor Rafael Galvão de Almeida, PhD, for his thoughtful guidance and constructive editorial oversight throughout the review process. We also acknowledge the invaluable contributions of our reviewers: Reviewer #1, Mohammed Salah, PhD, and Reviewer #2 (anonymous), whose detailed and insightful comments substantially improved the theoretical framework, methodological rigor, and overall clarity of our work. Their constructive criticism and scholarly suggestions were instrumental in strengthening the manuscript's contribution to the field.

## Author contributions

**Formal analysis:** Chao Tang.

**Methodology:** Chao Tang.

**Writing – original draft:** Chao Tang.

**Writing – review & editing:** Chao Tang.

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
