## [Decision Letter · Decision Letter 0]

24 Sep 2025

PONE-D-25-41907The Autonomy Paradox in AI-Generated Content Adoption: Extending the TAM Model in China’s Micro-Short Drama IndustryPLOS ONE

Dear Dr. Tang,

Thank you for submitting your manuscript to PLOS ONE. After careful consideration, we feel that it has merit but does not fully meet PLOS ONE’s publication criteria as it currently stands. Therefore, we invite you to submit a revised version of the manuscript that addresses the points raised during the review process.
The reviewers pointed issues with the manuscript, especially with its empirical part. You need to properly address these issues. Additionally, I can recommend ANATRASIRICHAI, N.; BULL, D. Artificial intelligence in the creative industries: a review. Artificial Intelligence Review, n. 55, p. 589-656, 2022 and HERTZMANN, A. Can computers create art? Arts, v. 7, n. 18, 2018. . It would help to situate better the use of AI in the creative process of microdramas. It should be noted that there must be a "quantitative" side to it. The proliferation of AI-generated "slop" has been hurting the arts in many sectors, not to mention the costs. It would be helpful if you expanded the concerns of AI usage in the meaning of the arts, how do creators deal with this issue?

We look forward to receiving your revised manuscript.

Kind regards,

Rafael Galvão de Almeida, PhD.

Academic Editor

PLOS ONE

3. In the online submission form you indicate that your data is not available for proprietary reasons and have provided a contact point for accessing this data. Please note that your current contact point is a co-author on this manuscript. According to our Data Policy, the contact point must not be an author on the manuscript and must be an institutional contact, ideally not an individual. Please revise your data statement to a non-author institutional point of contact, such as a data access or ethics committee, and send this to us via return email. Please also include contact information for the third party organization, and please include the full citation of where the data can be found.

5. We note that this data set consists of interview transcripts. Can you please confirm that all participants gave consent for interview transcript to be published?

If they DID provide consent for these transcripts to be published, please also confirm that the transcripts do not contain any potentially identifying information (or let us know if the participants consented to having their personal details published and made publicly available). We consider the following details to be identifying information:

- Names, nicknames, and initials

- Age more specific than round numbers

- GPS coordinates, physical addresses, IP addresses, email addresses

- Information in small sample sizes (e.g. 40 students from X class in X year at X university)

- Specific dates (e.g. visit dates, interview dates)

- ID numbers

Or, if the participants DID NOT provide consent for these transcripts to be published:

- Provide a de-identified version of the data or excerpts of interview responses

- Provide information regarding how these transcripts can be accessed by researchers who meet the criteria for access to confidential data, including:

a) the grounds for restriction

b) the name of the ethics committee, Institutional Review Board, or third-party organization that is imposing sharing restrictions on the data

c) a non-author, institutional point of contact that is able to field data access queries, in the interest of maintaining long-term data accessibility.

d) Any relevant data set names, URLs, DOIs, etc. that an independent researcher would need in order to request your minimal data set.

For further information on sharing data that contains sensitive participant information, please see: https://journals.plos.org/plosone/s/data-availability#loc-human-research-participant-data-and-other-sensitive-data

If there are ethical, legal, or third-party restrictions upon your dataset, you must provide all of the following details (https://journals.plos.org/plosone/s/data-availability#loc-acceptable-data-access-restrictions):

1. A complete description of the dataset

2. The nature of the restrictions upon the data (ethical, legal, or owned by a third party) and the reasoning behind them

3. The full name of the body imposing the restrictions upon your dataset (ethics committee, institution, data access committee, etc)

4. If the data are owned by a third party, confirmation of whether the authors received any special privileges in accessing the data that other researchers would not have

5. Direct, non-author contact information (preferably email) for the body imposing the restrictions upon the data, to which data access requests can be sent

Reviewers' comments:

Reviewer's Responses to Questions

**Comments to the Author**

1. Is the manuscript technically sound, and do the data support the conclusions?

Reviewer #1: Yes

Reviewer #2: Yes

2. Has the statistical analysis been performed appropriately and rigorously?

Reviewer #1: No

Reviewer #2: Yes

3. Have the authors made all data underlying the findings in their manuscript fully available?

Reviewer #1: No

Reviewer #2: Yes

4. Is the manuscript presented in an intelligible fashion and written in standard English?

Reviewer #1: Yes

Reviewer #2: Yes

5. Review Comments to the Author

Reviewer #1: Dear Authors,

Thanks for the submission. The topic is timely and the mixed-methods angle is promising. That said, there are a few core issues that need attention before the work is methodologically solid and ready for PLOS ONE.

Brief summary

You propose the Creative Industries Technology Acceptance Model (CITAM) to explain AIGC adoption among micro–short-drama creators (N=607), emphasizing innovation compatibility (IC) and a moderator you call creative autonomy retention (CAR). You combine CB-SEM with interviews and report support for all six hypotheses, including small but significant negative moderations.

What needs fixing (major)

Fit indices don’t line up.

You label the CFA fit “excellent” while reporting TLI = 0.85 (χ²(98)=179.729, CFI=0.988, RMSEA=0.037, SRMR=0.030). Later, the structural model shows TLI = 0.985. Please reconcile these numbers (likely 0.985 vs 0.85), report fit separately for measurement and structural models, and correct the interpretation.

The moderator’s identity shifts.

CAR is introduced as Creative Autonomy Retention, but a table note calls it Control and Autonomy Reactance. Those are not the same idea. Pick one definition and name, keep it consistent across text/tables/figures/appendix, clarify the direction of scoring (higher = what?), and state how any reverse-coded items were handled. Confirm unidimensionality.

Effect-size claims are off.

You cite f² ≈ 0.006 for the interaction(s) but say the moderation adds “~5% variance.” That doesn’t match. Please report the actual ΔR² from adding the interaction term, and tone down the claim if the gain is small. Also note whether predictors were centered and how the interaction was built (product term vs latent interaction).

Sampling description vs. achieved sample.

You describe purposive recruitment followed by stratified random sampling (with role targets like 40/30/20/10), yet the final role mix doesn’t match those targets. There’s also a “69% have 3+ years” statement that doesn’t align with the listed breakdown. Please clarify the sampling frame and execution, justify any representativeness claims (or drop them), and either weight post-hoc or be precise about the limits.

Data availability likely non-compliant.

PLOS ONE typically expects de-identified survey data in a public repository (with a DOI). Interviews can remain controlled-access with safeguards. As written, your “available on request via an ethics contact” approach for the survey dataset is unlikely to meet policy. Please deposit the de-identified survey data and codebook (e.g., OSF/Zenodo) or provide a policy-consistent exception.

Design label is inconsistent.

You call the design “explanatory sequential” and then describe a two-phase exploratory sequential (qual → quant). Choose the correct design, justify it, and align the narrative.

Positioning vs. TAM needs honesty.

You say you “extend TAM” but you drop PU/PEOU “for parsimony.” That reads less like an extension and more like an alternative tailored to creative contexts. Either reframe it as an alternative or include PU/PEOU as controls to show incremental value.

R² values drift.

R²(AUB) appears as 0.330 in one place and 0.335 elsewhere. Please make all R² and CI values consistent across text, tables, and figures.

Measurement transparency.

Keep HTMT, but please also share the Fornell–Larcker matrix and inter-construct correlations in the supplement, list all item loadings (flag any < .50 and how you handled them), and provide descriptive stats (M, SD, skew, kurtosis). If assumptions are borderline, consider robust ML (e.g., MLR).

Moderation/mediation reporting.

State clearly if you mean-centered variables, how the interaction was constructed, and whether you estimated latent interactions. Include simple-slope plots with CIs. For mediation, report the indirect effects with bootstrapped CIs and the number of draws, all in one consolidated results section.

Also fix (moderate/minor)

Terminology hygiene: Standardize the moderator’s name/abbreviation everywhere (text, tables, figure captions).

Causal tone: With cross-sectional data, avoid “influences/affects”; use “is associated with” or “predicts.”

Appendix completeness: Provide full item wording for IC, CAR, AI, and AUB, indicating any reverse-coded items; confirm CAR’s unidimensionality.

Common method bias: Harman’s single-factor test is not sufficient. Add a latent method factor or a marker-variable check and report the impact.

Tables/figures: Ensure the Table 1 note matches the construct names, and add a table of descriptive stats + correlations for all constructs.

Policy references: Where you mention policy implications (e.g., EU AI Act), add precise citations in the reference list.

Strengths worth keeping

Topical and practical contribution for creators and managers.

Mixed-methods design that helps interpret the small moderation effects.

Bottom line

Recommendation: Major revision. If you:

(1) fix the fit-index inconsistencies and right-size the effect-size claims,

(2) lock down the moderator’s definition/measurement,

(3) clean up the sampling story and any representativeness language, and

(4) bring the data-availability statement into line with PLOS ONE policy,

the paper will be much stronger and on firmer methodological ground.

Happy to take another look once these are addressed.

Reviewer #2: I appreciate the time and effort put into this research. From the article it is evident that this research has been carried out with high rigour, ensuring key concepts and hypotheses have been clarified, and supported with substantial literature review.

By saying that, I also find this paper to be too ambitious at moments, attempting to cover too much at once drawing from way far too many theories that only get partially explained. This is especially pronounced in Theoretical Framework and Hypotheses section (p.200 onwards) where many complex theories such as SDT, notion such as 'enhanced cognitive fit' (p.218), both asking for more explanation and evidence.

Same goes for 'creative identity threat theory [38] and the concept of psychological ownership [34], mentioned in p258.

This is again repeated in 'This defensive mechanism is also expected to impact actual usage. Drawing on

cognitive load theory [40] and identity protection mechanisms [41] (p270) - these concepts need more explanation, it is not enough to just say that the research is drawing upon them and cite one article. These are complex theories asking for substantial explanation. One approach could be to consider expanding on these theories through the final discussion in the paper, once the empirical data has been showed.

I believe that by expanding on these areas rather than just listing them, will make this research even stronger.

6. PLOS authors have the option to publish the peer review history of their article (what does this mean?). If published, this will include your full peer review and any attached files.

Reviewer #1: **Yes:** Mohammed Salah,PhD

Reviewer #2: No

---

## [Author Response · Author response to Decision Letter 1]

29 Sep 2025

Dear Dr. Rafael Galvão de Almeida and Esteemed Reviewers,

We would like to express our sincere gratitude for the invaluable feedback and guidance provided throughout the review process. The constructive comments and suggestions from both the reviewers and the editor have been instrumental in enhancing the quality of our manuscript.

We deeply appreciate Reviewer #1: Dr. Mohammed Salah for their meticulous evaluation and insightful suggestions regarding our statistical analysis and theoretical positioning. Your detailed feedback has significantly improved the clarity and rigor of our work.

We are equally grateful to Reviewer #2 for their thoughtful and thorough critique of our theoretical framework. Your emphasis on the need for deeper elaboration of complex theories has led to substantial improvements in our manuscript and its theoretical grounding.

We would also like to extend our special thanks to Editor Dr. Rafael Galvão de Almeida for your professional editorial guidance and for recommending key references that enriched our theoretical framework. Your advice has been invaluable in better situating our research within the broader discourse on AI in creative industries.

We are grateful for your time and effort in reviewing our manuscript, and we are confident that the revisions we have made, based on your expert advice, have substantially strengthened the manuscript.

Thank you once again for your support and for helping us improve our work.

Sincerely,

Chao Tang

Instructor

Department of Digital Media Arts

School of Animation

Jinling Institute of Technology

Nanjing, Jiangsu, China

Email: tangchao726@jit.edu.cn

---

## [Editor Report · Decision Letter 1]

22 Oct 2025

The autonomy paradox in AI-generated content adoption: creative-specific alternative to TAM model in China's micro-short drama industry

PONE-D-25-41907R1

Dear Dr. Tang,

We’re pleased to inform you that your manuscript has been judged scientifically suitable for publication and will be formally accepted for publication once it meets all outstanding technical requirements.

Kind regards,

Rafael Galvão de Almeida, PhD.

Academic Editor

PLOS ONE

Additional Editor Comments (optional):

The manuscript has some typos, please check it again.
---

## [Editor Report · Acceptance letter]

PONE-D-25-41907R1

PLOS One

Dear Dr. Tang,

I'm pleased to inform you that your manuscript has been deemed suitable for publication in PLOS One. Congratulations! Your manuscript is now being handed over to our production team.

Kind regards,

on behalf of

Dr. Rafael Galvão de Almeida

Academic Editor

PLOS One